# The Combinations of Physical Activity, Screen Time, and Sleep, and Their Associations with Self-Reported Physical Fitness in Children and Adolescents

**DOI:** 10.3390/ijerph19105783

**Published:** 2022-05-10

**Authors:** Zhenhuai Chen, Guijun Chi, Lei Wang, Sitong Chen, Jin Yan, Shihao Li

**Affiliations:** 1Faculty of Physical Education, China West Normal University, Nanchong 637001, China; czh208812@163.com; 2China Volleyball College, Beijing Sport University, Beijing 100084, China; chiguijun19830429@126.com; 3Department of Physical Education, Tangshan Normal University, Tangshan 063000, China; 4School of Physical Education and Sport Training, Shanghai 200438, China; wanglei@sus.edu.cn; 5Institute for Health and Sport, Victoria University, Melbourne, VIC 8001, Australia; sitongchen@szu.edu.cn; 6Centre for Active Living and Learning, University of Newcastle, Callaghan, NSW 2308, Australia; 7College of Human and Social Futures, University of Newcastle, Callaghan, NSW 2308, Australia; 8Department of Physical Education, China Agricultural University, Beijing 100083, China

**Keywords:** fitness promotion, movement guidelines, elementary school students, public health

## Abstract

Much evidence has indicated that adherence to the 24 h movement guidelines (physical activity, screen time and sleep) is associated with physical health, while little is known about the adherence to the 24 h movement guidelines and self-reported physical fitness in adolescents. This study, therefore, aims to explore the association between the 24 h movement guidelines (in isolation or combination) and self-reported physical fitness in a sample of Chinese adolescents in an age range of 10–17. Methods: A convenient sample of 3807 children and adolescents from 12 schools was adopted in the present study. A questionnaire based on the Health Behaviour in School-aged Children was used to assess physical activity and screen time, and the Pittsburgh Sleep Scale was utilized to measure sleep duration. The International Fitness Scale was used to assess physical fitness in study participants. Ordinal logistic regression was performed to estimate the association between adherence to the 24 h movement guidelines and self-reported physical fitness. Results: Of all study participants, 0.9% of them met the 24 h movement guidelines, and meeting the guidelines was significantly associated with higher levels of self-reported physical fitness. The analysis for the association between specific combinations of 24 h movement guidelines and self-reported physical fitness underscored the importance of meeting the physical activity recommendations. Conclusion: Adherence to more recommendations contained in the 24 h movement guidelines was associated with higher self-reported physical fitness, especially cardiorespiratory fitness and muscular strength. Our study also stressed the importance of promoting moderate to vigorous physical activity in children and adolescents. Further works should focus on the association of a recommendation of adherence with other health indicators and replicate this study on larger samples among Chinese children and adolescents. Additionally, longitudinal or interventional studies that include more socio-demographic factors are needed to explore the association between 24 h movement guidelines and self-reported physical fitness, and the 24 h movement guidelines also should be promoted on a large scale in Chinese children and adolescents. Moreover, it is also needed to gain better insights into the directionality of the relationship between compliance with 24 h movement guidelines and self-reported physical fitness, as well as the mechanisms underlying the associations in Chinese children and adolescents.

## 1. Introduction

Physical fitness is defined as a set of attributes that people have or achieve to maintain physical activity [1], and it is widely acknowledged that physical fitness is an important marker of health in children and adolescents [2,3]. Substantial evidence demonstrates the positive associations between enhanced physical fitness and various health indicators [4,5,6,7,8]. Physical fitness is considered as skill-related and health-related fitness, and as components of health-related physical fitness (HRPF) [9], that is, flexibility, muscular endurance, muscular strength, body composition, and cardiorespiratory fitness [10]. A number of pieces of evidence prove the benefits of high HRPF levels and the importance of health in adolescents and children [11,12], for example, if the adolescents have a greater level of HRPF, then they tend to have a reduced risk of cardiovascular disease in the future [13]. There is a direct association between cardiorespiratory fitness and life quality and mental well-being in adolescents [14]. Moreover, there is a positive association between greater cardiorespiratory fitness and academic performance [15]. There is a prospective association between the muscular factors, such as endurance and strength, and reduced risk of cardiometabolic parameters and adiposity, while a positive relationship is found between muscular fitness and bone health in future life [6,16]. Accordingly, improving physical fitness in children and adolescents is regarded as a significant factor in public health promotion.

Promoting physical activity, limiting excessive screen time, and encouraging sleep duration are meaningful approaches to adopt in improving physical fitness. In fact, there is a growing body of evidence to show the positive associations between sufficient physical activity [17,18], limited screen time [19,20,21,22], and appropriate sleep duration and higher levels of physical fitness in children and adolescents. This evidence calls for the adoption of holistic approaches to improve physical activity, screen time, and sleep (collectively referred to as 24 h movement behaviors) to enhance physical fitness, which can aid in the design and implementation of efficient fitness interventions.

Informed by a recent investigation, some health guidelines have reported on recommendations for physical activity, screen time, and sleep duration, leading to the recommendation of 60 min of moderate to vigorous physical activity (MVPA) per day [23,24], no more than 2 h of screen time per day, and age-specific sleep duration (9–11 h for children of 5–12 ages and 8–10 h for adolescents of 13–17 ages) for children and adolescents [23,25,26]. In line with this, some studies have explored the association between the 24 h movement guidelines and physical fitness in children and adolescents. For example, Carson et al. found that meeting these guidelines was associated with better physical fitness [25]. Some other studies have also suggested that meeting them is associated with physical fitness indicators [26,27]. However, evidence of contrary findings were reported in previous research [28].

In these studies, physical fitness was measured using field-based assessments, including a 20-m shuttle run for cardiorespiratory fitness and push-up tests for muscular strength. More recently, extending beyond these field-based assessments, researchers have developed a self-reported physical fitness assessment (International Fitness Scale [IFIS]) to evaluate perceptions of their fitness levels [29]. Evidence has indicated that self-reported physical fitness is positively associated with health outcomes in children and adolescents, demonstrating the validity and usefulness of the IFIS in health promotion research. The 24 h movement guidelines (recommend that ≥60 min of daily moderate-to-vigorous physical activity, ≤2 h of daily recreational screen time, 9–11 h and 8–10 h for nightly sleep duration for 6–13-year-olds and 14–17-year-olds, respectively) was initially developed by Canadian researchers, representing a paradigm shift in considering movement behaviors from a focus on a single specific movement behavior composition (e.g., MVPA) to an integrated movement behavior paradigm [30,31]. There is mixed evidence about the relationship between adiposity indicators (e.g., body fat percentage and body mass index), and 24 h movement guidelines in compliance with sleep, behavior, sedentary, and PA [27,32]. It is found by some studies that adolescents and children meeting these guidelines show low levels of body mass index (BMI) while those failing to meet all the guidelines show higher levels of BMI [33,34]. However, it is also indicated by some researchers that there is no relationship between BMI and guidelines compliance. 

Supporting the relationship between the 24 h movement guidelines and physical fitness, a cross-sectional study was published by Tanaka et al. [28] explored whether meeting the 24 h movement guidelines was associated with physical fitness levels in primary school children and found that (1) meeting all three 24 h movement guidelines was not associated with measures of physical fitness; (2) meeting the MVPA recommendation was associated with greater aerobic fitness and muscle endurance. While this study provided a valuable contribution to our current understanding of the relationship between the 24 h movement guidelines and physical fitness, the authors focused on the field-based assessment for physical fitness. Nonetheless, given that the effects of COVID-19 are still ongoing, some field-based fitness assessments cannot be undertaken in school settings. Consequently, there is a greater need to rely on self-reported physical fitness assessments. Thus, this study aims to explore the association between the 24 h movement guidelines and physical fitness assessed using a self-reported measure.

## 2. Methods

### 2.1. Participants and Procedure

This cross-sectional study was conducted to investigate the associations between physical activity, screen time, sleep, and self-reported physical fitness. This study was conducted between March and October in southeastern China in 12 public schools (5 elementary schools, 5 middle schools, and 2 high schools) located in 4 cities. Participating schools were contacted by the research staff. In each school, 1–3 classes of each grade were randomly selected by a contact assigned to each school. This procedure facilitated the recruitment of the initial sample consisting of 3807 children and adolescents. For the purposes of this study and in order to undertake further analysis, 2407 study participants were included, as they provided valid data on the required variables. All the children and adolescents involved in the study, as well as their parents or guardians, were specifically advised that participation was entirely voluntary. The study protocol and procedures were approved by the Institutional Review Board (IRB) of the Shanghai University of Sport and the grant number was 102772021RT071.

### 2.2. Measures

Analysis of physical activity, screen time, and sleep duration in this study was based on the Canadian 24 h Movement Guidelines for Youth and Children [35]. 

#### 2.2.1. Physical Activity

Physical activity was measured by one reliable and valid item derived from the Health Behaviour in School-Aged Children (HBSC) questionnaire [36]. Participants were required to answer the following question: “How many days did you engage in MVPA for at least 60 min on weekdays over the past week? (0 = none, 1 = 1 day, 2 = 2 days, 3 = 3 days, 4 = 4 days, 5 = 5 days, 6 = 6 days, and 7 = 7 days).” In order to provide a better understanding of MVPA, it was defined as “a variety of activities that can make your heartbeat faster and make you gasp for a while, including physical activities in physical education classes and daily life (such as running, playing ball)”. MVPA of more than 60 min per day is a clear threshold for assessing whether adolescents meet the Canadian 24-Hour Movement Guidelines for Youth and Children for its part of the MVPA recommendation [37]. 

#### 2.2.2. Screen Time

Screen time was measured based on reliable and valid items derived from the HBSC. Participants were required to answer the following questions: (1) How many hours did you spend watching TV or movies in your leisure time on weekdays and weekend days over the past week, respectively? (2) How many hours did you spend playing video games in your leisure time on weekdays and weekend days over the past week, respectively? (3) How many hours did you spend in activities using electronic screen-based devices in your leisure time on weekdays and weekend days over the past week, respectively? The responses to these questions could be none, approximately half an hour, 1 h, 2 h, 3 h, or more. According to the Canadian 24-H Movement Guidelines, screen time ≤ 2 h per day is deemed to meet the screen time guidelines [37]. 

#### 2.2.3. Sleep Duration

Sleep duration was measured using the Pittsburgh sleep quality index (PSQI) [38], which is a self-reported questionnaire to record the sleep quality of participants in the last month. Participants were required to answer the following questions: (1) During the past month, what time have you usually gone to bed at night? (2) During the past month, how long (in minutes) has it usually taken you to fall asleep each night? (3) During the past month, what time have you usually gotten up in the morning? The PSQI consists of seven components, including subjective sleep quality, sleep latency, sleep duration, habitual sleep efficiency, sleep disturbance, use of sleep medications, and daytime dysfunction. Previous studies have indicated that this scale has good reliability and validity among Chinese children and adolescents [37].

#### 2.2.4. Physical Fitness

The IFIS was used to evaluate self-estimations of physical fitness, using a 5-point Likert scale (very poor, poor, average, good, and very good). The IFIS contains five components, including general physical fitness, cardiorespiratory fitness, muscular strength, speed and agility, and flexibility. The following item was used: My general physical fitness is? (1 = very poor, 2 = poor, 3 = Average, 4 = Good, 5 = Very Good). The scale has reported acceptable reliability and validity in adolescents [29]. In addition, 544 Chinese adolescents were recruited to participate in a reliability study. Results indicated that the IFIS has acceptable reliability (weighted Kappa: 0.42–0.52 of the IFIS components and with a Cronbach’s alpha of 0.72) in Chinese children and adolescents [39].

#### 2.2.5. Socioeconomic Status

A number of sociodemographic variables were also gathered, including age, sex, siblings, whether living with parents or not, grade, residence, father’s and mother’s education levels, and perceived family affluence (0–10 scale). These variables were treated as covariates when performing further statistical analysis.

### 2.3. Statistical Analysis

Statistical analysis was carried out using the software program SPSS 26.0. Descriptive statistics (mean/standard deviation and percentage) were used to report on the basic characteristics of the study sample. Mean value with standard deviation was a continuous variable (e.g., age), whilst percentage was a categorical variable (e.g., grade, residence). Two sets of regression models were established: the first set was to determine the associations between a number of the 24 h movement guidelines met and physical fitness (five separate regression models); the second set was to determine the associations between combinations of the 24 h movement guidelines met and physical fitness (five separate regression models). A generalized linear model with ordinal logistic regression was used to achieve the association estimation. The statistical significance was set at *p* < 0.05.

## 3. Results

As presented in Table 1, the sample consisted of 52.7% boys and 47.3% girls, with a mean age recorded of 13.82 years (standard deviation = 2.1). In terms of the exposure variables, 6.5%, 43.2%, and 30.5% of participants met the MVPA guideline, the screen guideline, and the sleep guideline, respectively. Nearly half of the participants met one guideline, while only 0.9% of participants met all three guidelines. The proportion met by participants varied across specific combinations of 24 h movement guidelines, ranging from 0.9% (all) to 36.0% (none). Regarding the outcome variables, approximately half of the participants thought that they were average. Furthermore, a greater proportion of participants reported “good” and “very good” than “poor and very poor” on all outcome variables, with the exception of flexibility. 

Figure 1 presents the associations between the number of guidelines met and physical fitness indicators. As compared to meeting none of the guidelines, meeting two or more guidelines was significantly associated with better general physical fitness, cardiorespiratory fitness, and muscular strength. No significant associations were observed between the number of guidelines met and speed, agility, and flexibility. Additionally, a similar dose–effect pattern was observed, with the more guidelines being met corresponding with larger odds ratios (ORs).

Figure 2 shows the associations between the specific combinations of guidelines met and physical fitness indicators. On selecting meeting none of the guidelines as a reference point, meeting all, MVPA and sleep, MVPA and screen, and MVPA only were related to significantly better general physical fitness, cardiorespiratory fitness, and muscular strength; meeting all, MVPA and sleep, MVPA and screen, sleep only, and MVPA only were correlated with significantly better speed and agility; and meeting MVPA and screen and MVPA only were significantly correlated with better flexibility.

## 4. Discussion

The main aim of the current study is to examine the association between 24 h movement guidelines and self-reported physical fitness indicators (e.g., overall fitness, cardiorespiratory fitness) in children and adolescents. The results revealed that meeting the 24 h movement guidelines was associated with higher levels of physical fitness indicators. Furthermore, this association may primarily be dependent on meeting the physical activity guideline. This study also provided evidence that screen time and sleep duration may not be key contributors to improved physical fitness in children and adolescents. Finally, extremely few children and adolescents reported an optimal combination of physical activity, screen time, and sleep duration. 

Only 0.9% of children and adolescents met the 24 h movement guidelines in this study. This finding should be a cause of concern and prioritized attention for relevant researchers and health practitioners, as the combinations of physical activity, screen time, and sleep are responsible for promoting health among children and adolescents [40,41,42]. In light of this result (0.9%), which was extremely lower than other previous studies [31,43], measures are needed to tackle this public health issue. 

The results of the current study indicate that meeting the 24 h movement guidelines was more likely to lead to higher levels of self-reported physical fitness. This finding is consistent with some earlier research [25]. The current study findings clearly demonstrate that sufficient physical activity and limited screen time both correlate with improved physical fitness [4,5,18]. Nevertheless, these findings do not concur with a study based on a Japanese sample [28]. One possible explanation accounting for this discrepancy may be due to the different measurements it used in contrast to the current study. Undertaken in Japan, Hui et al. study applied device-based measures to assess levels of physical activity [28], along with different measurements to assess physical fitness. In general, the current study advances knowledge in the area of 24 h movement behavior research, which can assist in updating and refining the guidelines in the future. However, such evidence remains limited, thus highlighting the need for additional empirical studies to be conducted to re-examine the present research findings.

Interestingly, further investigation of the combinations of the 24 h movement guidelines and physical fitness indicators revealed that meeting the physical activity guidelines was generally significantly associated with all self-reported physical fitness indicators. Some studies support the findings that screen time and sleep duration are not correlated with physical fitness indicators in children and adolescents. In addition to this supportive evidence, studies using compositional data analysis have highlighted that encouraging physical activity, especially that of moderate to vigorous intensity levels, can lead to better outcomes in terms of promoting physical fitness [44,45,46]. This may indicate that the importance of limited screen time and sleep duration is weaker (even null) than sufficient physical activity in the promotion of physical fitness. Evidence exists of the role MVPA can potentially play in eliminating the adverse health effects of excessive sedentary behavior (including screen time). Collectively, this analysis can help to explain the function physical activity, as opposed to screen time and sleep duration, and can perform better in terms of higher levels of physical fitness.

Based on these research results (see Figure 2), meeting either screen time or sleep guidelines was not associated with any physical fitness indicators in children and adolescents. Mixed research findings exist in relation to the association between screen time and physical fitness in children and adolescents. For example, there is evidence to indicate a negative association between screen time and physical fitness [19,20,21,47], which is inconsistent with the present study. On the contrary, some studies suggest that no significant association exists between screen time and physical fitness [48,49,50]. The discrepancy is largely attributable to the application of different measurement protocols in relation to physical fitness and movement behaviors. However, the current study used self-reported physical fitness measures, which to the authors’ knowledge is the first of its kind and which future studies are encouraged to replicate.

Another finding of the current study suggests that there is no association between meeting the sleep guideline and physical fitness. However, this finding appears to be controversial, given that some studies have put forward different arguments. Loprinzi and Joyner [51] indicated a positive association between meeting sleeping guidelines and health-related quality of life (HRQOL), including indicators such as body mass index (BMI), blood pressure, and diabetes. However, the study sample undertaken by Loprinzi and Joyner [51] was drawn from a random selection of United States citizens, with an average age of 46.8 years. Therefore, the latter study might not adequately refute the findings of the current research. Furthermore, while Kracht, Champagne [52] generally acknowledge the association between meeting sleep guidelines and physical fitness, they contended that sleep exerts less impact on physical health than physical activity and dietary habits. They also suggested that no relationship exists between sleep and body fat percentage. Therefore, the association between meeting sleep guidelines and physical fitness warrants further investigation in the future.

Study Limitations and Strengths

This study has some inherent limitations in terms of study design, measures, and participants. Reliance on a cross-sectional study design resulted in failing to draw any causal conclusions. Accordingly, the directionality of the association between meeting 24 h movement guidelines and physical fitness indicators cannot be determined. Second, this study utilized self-reported measures to collect data on all the variables, which was subject to recall bias and social desirability of study participants. Finally, as this study adopted a convenient and non-probability sampling method, research findings may be regionally as opposed to nationally replicable. Given these points, future studies are encouraged to address these limitations in order to generate stronger evidence. However, despite these limitations, this study contains a number of strengths. To the best of the authors’ knowledge, this study is the very first of its kind to date to assess the associations between meeting 24 h movement guidelines and self-reported physical fitness indicators, thus contributing to and extending the current body of literature. Furthermore, as the sample size was large, sufficient statistical power could be reached. Finally, this study was controlled for multiple covariates to ensure the accurate estimation of the association between meeting 24 h movement guidelines and self-reported physical fitness. 

As a result, 24 h movement guidelines specify different physical activity intensities (vigorous, moderate and light), sleep and sedentary behaviors. With the guidelines, professionals engaging in health care or public health, educators, parents, youths, children, or government can know the importance of different movement behaviors within a day. Moreover, we can also have an agenda for healthy active life by referring to the 24-H Movement Behaviour Guidelines, which can contribute to the improvement in general well-being and health among youth and children in China.

## 5. Conclusions and Future Implications

In conclusion, meeting the 24 h movement guidelines was positively associated with higher levels of physical fitness in children and adolescents. It is noteworthy that this positive association was largely dependent on MVPA, as opposed to screen time and sleep duration. Future studies are encouraged to replicate this work so as to confirm or negate these research findings. The current research findings clearly highlight the health significance of MVPA as a behavioral and health priority in children and adolescents. 

In future studies, more work should investigate the relationships between 24 h movement guidelines and more other health indicators and replicate this study on larger samples in Chinese children and adolescents. Moreover, we should also pay more attention to the association between 24 h movement guidelines and self-reported physical fitness with a longitudinal study that controls for more socio-demographic factors. Finally, from a clinical perspective, it is urgent to promote positive 24 h movement behaviors in Chinese children and adolescents, as it would improve their additional health-related outcomes. Moreover, based on the research findings, it is promising that increasing MVPA could be a more significant correlate with 24 h movement behavior research. 

## Figures and Tables

**Figure 1 ijerph-19-05783-f001:**
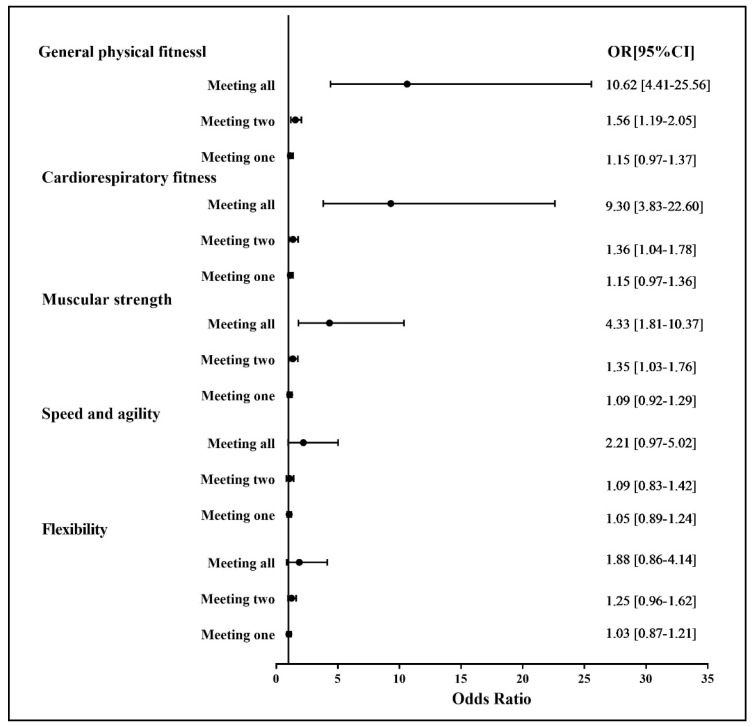
Association between 24 h movement guidelines (number met) and self-reported physical fitness indicators. OR: odds ratio; CI: confidence interval. Reference group: meeting none.

**Figure 2 ijerph-19-05783-f002:**
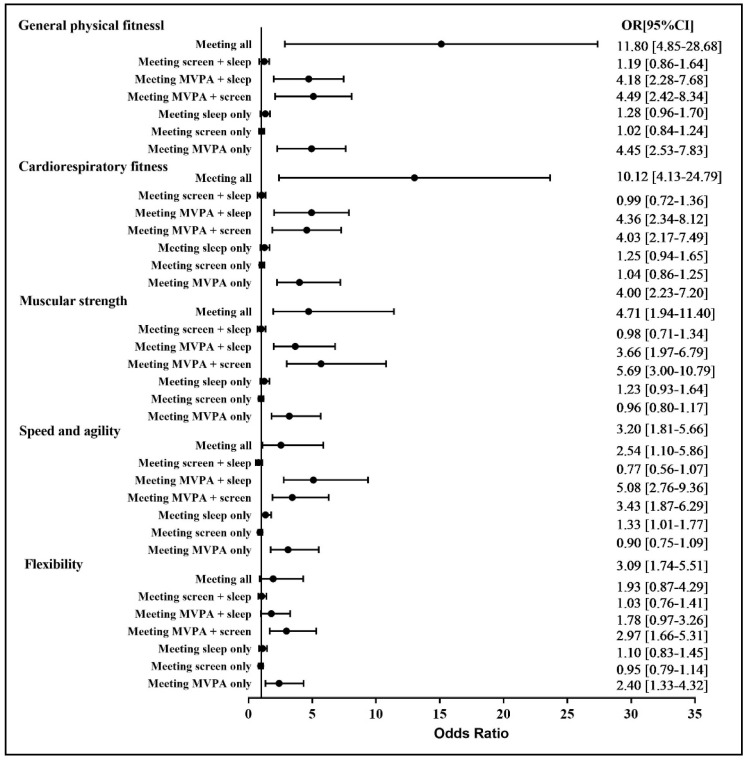
Association between 24 h movement guidelines (specific combinations) and self-reported physical fitness indicators. OR: odds ratio; CI: confidence interval. Reference group: meeting none.

**Table 1 ijerph-19-05783-t001:** Sample characteristics of this study.

Variables		n/Mean	%/SD
Age		13.82	2.1
Gender			
	Boy	1268	52.7
	Girl	1139	47.3
Siblings			
	Yes	1184	49.2
	No	1223	50.8
Living with parent			
	Yes	2018	83.8
	No	389	16.2
Grade			
	4	355	14.7
	5	334	13.9
	7	353	14.7
	8	415	17.2
	10	515	21.4
	11	435	18.1
Residence			
	Rural	277	11.5
	Suburban	526	21.9
	Urban	1604	66.6
Father Education Level			
	Middle school or below	623	25.9
	High school	595	24.7
	Undergraduate	772	32.1
	Graduate	137	5.7
	Unknown	280	11.6
Mother Education Level			
	Middle school or below	770	32.0
	High school	512	21.3
	Undergraduate	732	30.4
	Graduate	114	4.7
	Unknown	279	11.6
Family affluence		5.09	1.5
MVPA guideline			
	Not meet	2250	93.5
	Meet	157	6.5
Screen guideline			
	Not meet	1368	56.8
	Meet	1039	43.2
Sleep guideline			
	Not meet	1674	69.5
	Meet	733	30.5
24 h movement guideline number			
	None	866	36.0
	One	1175	48.8
	Two	344	14.3
	Three	22	0.9
24 h movement guideline combinations			
	None	866	36.0
	MVPA only	44	1.8
	Screen only	718	29.8
	Sleep only	413	17.2
	MVPA + screen	46	1.9
	MVPA + sleep	45	1.9
	Screen + sleep	253	10.5
	All	22	0.9
General physical fitness			
	Very poor	78	3.2
	Poor	297	12.3
	Average	1254	52.1
	Good	581	24.1
	Very good	197	8.2
Cardiorespiratory fitness			
	Very poor	90	3.7
	Poor	341	14.2
	Average	1140	47.4
	Good	621	25.8
	Very good	215	8.9
Muscular strength			
	Very poor	85	3.5
	Poor	380	15.8
	Average	1239	51.5
	Good	549	22.8
	Very good	154	6.4
Speed/Agility			
	Very poor	63	2.6
	Poor	305	12.7
	Average	1120	46.5
	Good	667	27.7
	Very good	252	10.5
Flexibility			
	Very poor	166	6.9
	Poor	530	22.0
	Average	1031	42.8
	Good	491	20.4
	Very good	189	7.9

Note. SD: standard deviation.

## Data Availability

The data presented in this study are available on request from the corresponding author.

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
