# Peer review of "The Combinations of Physical Activity, Screen Time, and Sleep, and Their Associations with Self-Reported Physical Fitness in Children and Adolescents"

_ijerph, 2022, doi:10.3390/ijerph19105783_

Round 1

Reviewer 1 Report

Thank you for the Authors contribution to this manuscript. In my general opinion, it is an interesting topic, and the study design is appropriate, but there is a lack of information in the introduction and the methods that make the study hard to understand.  

I have the following suggestions for the Authors:

  • The main goal of this study is "to explore the association between the 24-hour movement guidelines and physical fitness assessed using a self-reported measure." (Line 74-75), but we have little information on these associations in the introduction. I recommend expanding the first part with more studies.
  • It's not clear what 24-hour movement behaviors mean. Please define.
  • Please define physical fitness as well.
  • The Introduction is generally short.
  • Methods chapter should be divided. I have the following recommendations: 2.1. Participants and Procedure; 2.2. Measures (divided into 2.2.1 etc. for every questionnaire) 2.3. Statistical analysis.
  • Please pay attention to the descriptions of the measures. Include how many items were used what was the answer categories if you can please give examples.
  • It's not clear how the 24-hour movement guideline was combinate
  • The Authors present the IFIS as the frequencies of the answer categories. However, it seems it has a mean value as well. I believe the means could represent the sample better on cardiorespiratory fitness, muscular strength, etc.
  • Please be aware that MVPA, Physical Activity, and Physical Fitness have different meanings. Please use these terms carefully.

Author Response

Responses to Reviewer 1 Comments

Dear reviewers,

Thank you for your time and valuable comments. We have provided a point-by-point response to each of your comments and suggestions and have made the appropriate changes to the manuscript. We believe the paper has improved significantly because of the review process.

Response to Reviewer comments

Title: The combinations of physical activity, screen time, and sleep, and their associations with self-reported physical fitness in children and adolescents

Manuscript reference: ijerph-1647221

Reviewer comments in bold type

Response in normal type

Amended text in italics

Reviewer 1

I have the following suggestions for the Authors:

The main goal of this study is "to explore the association between the 24-hour movement guidelines and physical fitness assessed using a self-reported measure." (Line 74-75), but we have little information on these associations in the introduction. I recommend expanding the first part with more studies.

Response: Thank you for suggestion, please see line 90-116, The manuscript now reads:

The 24-h movement guidelines (recommend that ≥60 min of daily moderate-to-vigorous physical activity, ≤2 h of daily recreational screen time, 9-11h and 8-10h for nightly sleep duration for 6-13-year-olds and 14-17-year-olds, respectively) was initially developed by Canadian researchers, representing a paradigm shift in considering movement behaviors from a focus on a single specific movement behavior composition (e.g., MVPA) to an integrated movement behaviour paradigm [24]. There is mixed evidence about the relationship between adiposity indicators (eg. Body fat percentage and body mass index), and 24-hour movement guidelines compliance for sleep, behaviour, sedentary and PA [34, 35]. It is found by some studies that, adolescents and children meeting these guidelines show low levels of body mass index (BMI) while those failing to meet all the guidelines show higher levels of BMI [36, 37]. However, it is also indicated by some researchers that there is no relationship between BMI and guidelines compliance.

Supporting the relationship between the 24-hour movement guidelines and physical fitness, a cross-sectional study was published by Tanaka et al [38] explored whether meeting the 24 h movement guidelines was associated with physical fitness levels in primary school children and found that 1) meeting all three 24 h movement guidelines was not associated with measures of physical fitness; 2) meeting the MVPA recommendation was associated with greater aerobic fitness and muscle endurance. While this study provided a valuable contribution to our current understanding of the relationship between the 24-hour movement guidelines and physical fitness, the authors focused on the field-based assessment for physical fitness. Nonetheless, given that the effects of COVID-19 are still ongoing, some field-based fitness assessments cannot be undertaken in school settings. Consequently, there is a greater need to rely on self-reported physical fitness assessments. Thus, this study aims to explore the association between the 24-hour movement guidelines and physical fitness assessed using a self-reported measure.

It's not clear what 24-hour movement behaviours mean. Please define.

Response: Thank you for suggestion, please see line 90-95, The manuscript now reads:

The 24-h movement guidelines (recommend that ≥60 min of daily moderate-to-vigorous physical activity, ≤2 h of daily recreational screen time, 9-11h and 8-10h for nightly sleep duration for 6-13-year-olds and 14-17-year-olds, respectively) was initially developed by Canadian researchers, representing a paradigm shift in considering movement behaviors from a focus on a single specific movement behavior composition (e.g., MVPA) to an integrated movement behaviour paradigm [32, 33].

Please define physical fitness as well.

Response: Thank you for suggestion, please see line 46-47, The manuscript now reads:

Physical fitness is defined as a set of attributes that people have or achieve to maintaining physical activity [1],.........

The Introduction is generally short.

Response: Thank you for suggestion, we have expanded the introduction. Please see line 50-62, 96-102.

Physical fitness is considered skill-related and health-related fitness, and there are components of health-related physical fitness (HRPF) [9], that is, flexibility, muscular endurance, muscular strength, body composition and cardiorespiratory fitness [10]. A number of pieces of evidence prove the benefits of high HRPF levels and the importance of health in adolescents and children [11, 12], for example, if the adolescents have a greater level of HRPF, then he tends to have a reduced risk of cardiovascular disease in the future [13]. There is a direct association between cardiorespiratory fitness and life quality & mental well-being in adolescents [14]. Moreover, there is a positive association between greater cardiorespiratory fitness and academic performance [15]. There is a prospective association between the muscular such as endurance and strength and reduced risk of cardiometabolic parameters and adiposity, while a positive relationship is found between muscular fitness and bone health in future life [6, 16].

There is mixed evidence about the relationship between adiposity indicators (eg. Body fat percentage and body mass index), and 24-hour movement guidelines compliance for sleep, behaviour, sedentary and PA [34, 35]. It is found by some studies that, adolescents and children meeting these guidelines show low levels of body mass index (BMI) while those failing to meet all the guidelines show higher levels of BMI [36, 37]. However, it is also indicated by some researchers that there is no relationship between BMI and guidelines compliance.

The methods chapter should be divided. I have the following recommendations: 2.1. Participants and Procedure; 2.2. Measures (divided into 2.2.1 etc. for every questionnaire) 2.3. Statistical analysis.

Response: Thank you for suggestion, please see line 135, 147, 159, 170 and 181.

Please pay attention to the descriptions of the measures. Include how many items were used and what was the answer categories if you can please give examples.

Response: Thank you for suggestion, please find line 137-140, 149-155, 162-165, 171-176. The manuscript now reads:

Participants were required to answer the following question: “How many days did you engage in MVPA for at least 60 minutes on weekdays over the past week? (0 = none, 1 = 1 day, 2 = 2 days, 3 = 3 days, 4 = 4 days, 5 = 5 days, 6 = 6 days, and 7 = 7 days).”

  • How many hours did you spend watching TV or movies in your leisure time on weekdays and weekend days over the past week, respectively? (2) How many hours did you spend playing video games in your leisure time on weekdays and weekend days over the past week, respectively? (3) How many hours did you spend in activities using electronic screen-based devices in your leisure time on weekdays and weekend days over the past week, respectively?

Participants were required to answer the following questions: (1) During the past month, what time have you usually gone to bed at night? (2) During the past month, how long (in minutes) has it usually taken you to fall asleep each night? (3) During the past month, what time have you usually gotten up in the morning?

The IFIS was used to evaluate self-estimations of physical fitness, using a 5-point Likert scale (very poor, poor, average, good, and very good). The IFIS contains five components, including general physical fitness, cardiorespiratory fitness, muscular strength, speed and agility, and flexibility. The following item was used: My general physical fitness is? (1=very poor, 2=poor, 3=Average, 4=Good, 5=Very Good).

It's not clear how the 24-hour movement guideline was combinate

Response: Thank you for suggestion, please see line 90-96, The manuscript now reads:

The 24-h movement guidelines (recommend that ≥60 min of daily moderate-to-vigorous physical activity, ≤2 h of daily recreational screen time, 9-11h and 8-10h for nightly sleep duration for 6-13-year-olds and 14-17-year-olds, respectively) was initially developed by Canadian researchers, representing a paradigm shift in considering movement behaviors from a focus on a single specific movement behavior composition (e.g., MVPA) to an integrated movement behaviour paradigm [24].

The Authors present the IFIS as the frequencies of the answer categories. However, it seems it has a mean value as well. I believe the means could represent the sample better on cardiorespiratory fitness, muscular strength, etc.

Please be aware that MVPA, Physical Activity, and Physical Fitness have different meanings. Please use these terms carefully.

Response: Yes, we agree with that. However, the codes of IFIS are not actually continuous variables, it takes ordinal variables to present levels of assessment of IFIS. In this regard, using the ordinal variable in our study, we think, was more appropriate to present our results. May we request you can understand our explanation. Thank you very much.

Reviewer 2 Report

Physical activity and the promotion of a healthy lifestyle play an important role in the development of children. The introduction is generally very well. The study presents interesting findings and the results are processed at an excellent level. I recommend it for publication, but I suggest some changes.

In abstract change „convenient sample consisting of 12 school adolescents “

The conclusion in the abstract needs to be changed

The keywords‘ holistic public health“ needs to be changed.

change the word holistic throughout the manuscript,

No mentions of the aim in the discussion "The main aim of the current study is to examine the association between 24-hour movement guidelines and self-reported physical fitness indicators (e.g., overall fitness, cardiorespiratory fitness) in children and adolescents."

The study by Hui et al. applied without ’s

Author Response

Reviewer 2:

Physical activity and the promotion of a healthy lifestyle play an important role in the development of children. The introduction is generally very well. The study presents interesting findings and the results are processed at an excellent level. I recommend it for publication, but I suggest some changes.

In abstract change „convenient sample consisting of 12 school adolescents “

Response: Thank you for suggestion, please see line 20-21, The manuscript now reads:

A convenient sample of 3,807 children and adolescents from 12 schools was adopted in the present study.

The conclusion in the abstract needs to be changed

Response: Thank you for the suggestion, please see line 34-42, The manuscript now reads:

Further works should focus on the association of recommendation adherence with other health indicators and replicate this study on larger samples among Chinese children and adolescents. Besides, longitudinal or interventional studies that include more socio-demographic factors are needed to explore the association between 24-hour movement guidelines and self-reported physical fitness, and the 24-hour movement guidelines also should be promoted on a large scale in Chinese children and adolescents. Moreover, it is also needed to gain better insights into the directionality of the relationship between compliance with 24-hour movement guidelines and self-reported physical fitness, as well as the mechanisms underlying the associations in Chinese children and adolescents.

The keywords‘ holistic public health“ needs to be changed.

Response: Thank you for the suggestion, please see line 43.

change the word holistic throughout the manuscript,

Response: Thank you for the suggestion, please see line 72. The manuscript now reads:

Informed by a recent investigation

No mentions of the aim in the discussion "The main aim of the current study is to examine the association between 24-hour movement guidelines and self-reported physical fitness indicators (e.g., overall fitness, cardiorespiratory fitness) in children and adolescents."

Response: Thank you for the suggestion, our discussion is focusing on the interpretation of the relationship between 24-hour movement guidelines and self-reported physical fitness. Please find line 234-236.

The study by Hui et al. applied without ’s.

Response: Thank you for the suggestion, please see line 256. The manuscript now reads:

Hui et al. study applied

Reviewer 3 Report

The abstract and the keywords are clear enough and follow the contents of the paper. Overall, the paper presents a logical order of ideas about an interesting topic. The introduction is organized from the general to the particular, culminating in the formulation of objectives. The author presents a critical and sufficiently broad review of the existing theoretical-empirical evidence. The state of the art supports the problem. Limitations were adequately pointed out in the paper. The quality of the written and readability of the paper in English is fine. Needs punctuation improvement.

The sample and sample selection are adequately described. Despite the sampling process and its representativeness being reported as a strength the sampling process and its representativeness are not stated (it is suggested to add the power analysis or the power test). The instruments and procedures used are properly reported. The methodological design allows for investigating/testing the formulated Objectives. The quality of the methods used is guaranteed, and the ethical principles are safeguarded. The data analysis options are adequate and properly described.

The results seem credible and consistent with the title and objectives. The results are properly presented and the presentation of data and results is clear. Tables and figures are well-designed, relevant, and self-readable, allowing for easy reading and analysis.

The paper explores the association between the 24-hour movement guidelines (in isolation or combination) and self-reported physical fitness in a sample of Chinese adolescents. Despite the range of limitations listed by the authors, the discussion is pertinent, supported by the state of the art. The results and their implications/applications should be discussed in the broader context.

The conclusions follow from the presented results and systematize the central ideas of the work. The findings are presented as conclusions. We suggest that future research and lines of research should be better listed and highlighted. The references are relevant to the text.

Author Response

Reviewer 3:

The abstract and the keywords are clear enough and follow the contents of the paper. Overall, the paper presents a logical order of ideas about an interesting topic. The introduction is organized from the general to the particular, culminating in the formulation of objectives. The author presents a critical and sufficiently broad review of the existing theoretical-empirical evidence. The state of the art supports the problem. Limitations were adequately pointed out in the paper. The quality of the written and readability of the paper in English is fine. Needs punctuation improvement.

Response: Thank you very much!

The sample and sample selection are adequately described. Despite the sampling process and its representativeness being reported as a strength the sampling process and its representativeness are not stated (it is suggested to add the power analysis or the power test). The instruments and procedures used are properly reported. The methodological design allows for investigating/testing the formulated Objectives. The quality of the methods used is guaranteed, and the ethical principles are safeguarded. The data analysis options are adequate and properly described.

Response: Thank you very much!

The results seem credible and consistent with the title and objectives. The results are properly presented and the presentation of data and results is clear. Tables and figures are well-designed, relevant, and self-readable, allowing for easy reading and analysis.

Response: Thank you very much!

The paper explores the association between the 24-hour movement guidelines (in isolation or combination) and self-reported physical fitness in a sample of Chinese adolescents. Despite the range of limitations listed by the authors, the discussion is pertinent, supported by the state of the art. The results and their implications/applications should be discussed in the broader context.

Response: Thank you for suggestion, please see line 319--325, The manuscript now reads:

24 Hour Movement Behaviour Guidelines specify different physical activity intensities (vigorous, moderate and light), sleep and sedentary behaviours. With the guidelines, professionals engaging in health care or public health, educators, parents, youths, children or government can know the importance of different movement behaviours within a day. Moreover, we can also have an agenda for healthy active life by referring to the 24 Hour Movement Behaviour Guidelines, which contribution could be made to the improvement in general well-being and health among youth and children in China.

The conclusions follow from the presented results and systematize the central ideas of the work. The findings are presented as conclusions. We suggest that future research and lines of research should be better listed and highlighted. The references are relevant to the text.

Response: Thank you for the suggestion, please see line 334--343, The manuscript now reads:

In future studies, more work should investigate the relationships between 24hour movement guidelines and other health indicators and replicate this study on larger samples of Chinese children and adolescents. Moreover, we should also pay more attention to the association between 24-hour movement guidelines and self-reported physical fitness with a longitudinal study that controls for more socio-demographic factors. Finally, from a clinical perspective, it is urgently to promote positive 24-h movement behaviors in Chinese children and adolescents, as it would improve their additional health-related outcomes. Moreover, based on the research findings, it is promising to increase MVPA could be a more significant correlate in the 24hour movement behaviour research.

Round 2

Reviewer 1 Report

Thank you for the author's contribution! The manuscript significantly changed. I recommend publishing.